# Meshing Stiffness Calculation of Disposable Harmonic Drive under Full Load

**Yuxin Zhang** [1], **Xudong Pan** [1], **Yuefeng Li** [1,*], **Guanglin Wang** [1] **and Guicheng Wu** [2]

[1] School of Mechatronics Engineering, Harbin Institute of Technology, Harbin 150001, China; yuxinz@hit.edu.cn (Y.Z.); pxd@hit.edu.cn (X.P.); glwang@hit.edu.cn (G.W.)

[2] Sichuan Aerospace Fenghuo Servo Control Technology Co., Ltd., Chengdu 611130, China; wgc19831208@163.com

\* Correspondence: yuefengli@hit.edu.cn

**Abstract:** Mechanical equipment in the field of aerospace that is used only once is called disposable machinery. As a piece of typical disposable machinery, disposable harmonic gear exhibit stiffness failure with a large load. This manuscript distinguishes disposable harmonic gear from conventional harmonic gear in terms of the application environment and structure. Then, this paper determines the single-tooth stiffness of the disposable harmonic gear under full load by using the non-uniform beam model and the improved energy method. In addition, the multi-tooth meshing in the disposable harmonic drive is considered, and the improved energy method is modified. Besides, the normal contact force and comprehensive elastic displacement at each meshing position are calculated according to the finite element model. Additionally, curves of the single-tooth stiffness and the comprehensive meshing stiffness are obtained. The theoretical results of the modified analytical method and FEM are compared to verify the correctness of the proposed method in terms of calculating the meshing stiffness of the disposable harmonic drive. Finally, FEM is used to obtain the failure form of the disposable harmonic gear under overload.

**Keywords:** disposable harmonic drive; meshing stiffness; modified improved energy method; finite element model

## 1. Introduction

Different from conventional long-running and reused machinery, machinery which is not reused is named disposable machinery. Such machines are designed to have a very short service life (measured in "minutes") and take the use of a short-term high load as the working normality. The harmonic reducer in the disposable electromechanical actuator is a key component that determines the performance of the system. Harmonic gears transmit motion and powers through deformation waves caused by flexible parts with controllable deformation. Harmonic drives (HDs) have the advantages of a large transmission ratio, high transmission accuracy, and small volume weight, which are of great significance for the application of a high power–weight ratio [1]. The short-term extreme load limitation and dynamic properties of the disposable harmonic gear transmission have gradually developed into a focus of research in this area [2]. Due to the extremely short service life, the flexible wheel's high-cycle fatigue failure, which is the main concern in the research of conventional harmonic gears, will not appear in the disposable harmonic gear. Therefore, determining the meshing stiffness of harmonic gears is an important direction of research on full-load harmonic drives.

Many papers have discussed the meshing stiffness associated with gear transmission. The harmonic drive is characterized by a straight tooth profile, extremely thin rim, and simultaneous meshing of multiple teeth. The research findings on the meshing stiffness of cylinder gears with a thin rim and high engagement ratio can be used for reference. Most research on the stiffness of cylindrical gear transmission to date has used the analytical

method (AM) or the finite element method (FEM). As early as 1987, Yang et al. [3] proposed decomposing the total energy of gear meshing into Hertz contact energy, bending energy, and axial compression energy. The subsequent researches on the meshing stiffness of gear transmission were carried out on the conclusions of Yang. Considering the shear energy generated by the component of the contact force on the teeth of the gear, Tian [4] introduced shear stiffness using Yang's model. Based on the models proposed by Weber [5], Attia [6], and Cornell [7], Sainsot et al. [8] calculated the offset caused by the action of the teeth of the gear when its foundation is subjected to a force. Fakher [9] introduced the offset energy and the offset stiffness to calculate the stiffness of the gear meshing based on Sainsot et al.'s research. Sun et al. [10] divided the spur gear into thin slices along the width of the tooth and modified the model of meshing stiffness by considering the influences of lead crowning relief and tip relief. Wang et al. [11] equated contact among the teeth with the elastic contact of a spring to study the influence of the width of the web, its hole radius, and the length of the crack on the TVMS of a spur gear with webbing according to the potential energy method. This method could help to analyze the meshing stiffness when the gear foundation is shared under multi-tooth. Considering the periodically varying load distribution in tooth surface wear, Chen et al. [12] established a new model for calculating the TVMS of external spur gears. The model also discussed the effect of surface wear on stiffness. Sánchez et al. [13] studied the load distribution and meshing stiffness of standard and high-contact-ratio spur gears after profile modifications. The finite element method is also commonly used to solve the problems of tooth deformation and meshing stiffness of gear drives by employing finite element analysis software. Ma et al. [14] used ANSYS to establish a FEM of a cracked spur gear transmission to analyze the influence of extended tooth contact on the meshing stiffness owing to flexible teeth. This model could analyze the meshing stiffness of multi-tooth meshing under high torque, which provided help for the establishment of FEM in this paper. Based on the Quasi-static Algorithm (QSA), Zhan et al. [15] proposed an integrated CAD–FEM–QSA system to analyze the TVMS of gears. Compared with traditional methods, this technique had higher precision and efficiency. Chen et al. [16] proposed a FEM of the meshing stiffness of a spur gear by considering complex paths for gear and crack propagation based on finite element theory and the contact analysis of loaded teeth. The model proved that the meshing stiffness is affected by the rim thickness. The relationship between the meshing stiffness of the gear and the thickness of its web and fracture mode was studied. Considering that the FEM is less efficient at solving the TVMS of gears than the AM, and the results are affected by such factors as the division of the meshing, the FEM is generally used as a supplement to the AM.

Most of the analytical methods mentioned above focus on the meshing stiffness of spur cylindrical gears. The above methods and models confirmed that the meshing stiffness is affected by the thickness of the rim and the number of meshing teeth, which established the foundation for the research on the mesh stiffness of harmonic drive in this paper. In general, the flexible wheel in HD is prone to fatigue failure, because of which less research has been conducted on the stiffness of large-load flexible wheels. Considering the application environment and the extremely thin rim structure, it is necessary to study the stiffness of the transmission of a disposable harmonic gear. Gear deformation is the research basis for the meshing stiffness of the harmonic drive, and flexible wheel deformation occupies an important part of the total deformation. Ma et al. [17] discussed the flexible wheel deformation characteristics in HD under different driving speeds. Dong simplified the flexible wheel and obtained the strain and stress on the front, middle, and rear sections of the flexible rim [18]. Based on the finite element model, Kayabasi [19] analyzed the maximum stress and position of the flexible wheel during transmission so that the flexible tooth profile could be optimized. On the basis of the involute tooth profile optimization method proposed by Dong [20], Chen et al. [21] combined the variable section beam element and shell element. Then, ANSYS was used to analyze the deformation of the flexible teeth and neutral layer after assembly and transmission. In addition, they also studied the influence

of wave generators with different shapes on the flexible middle plane's deformation [22,23]. The variable-section beam and shell models mentioned in the above studies are common methods for the calculation of the deformation and stiffness of the flexible wheel. By analyzing the global sensitivity of the harmonic drive, Hrcek et al. [24] discussed the effects of geometric parameters, such as the module, tooth height, and rim thickness, on lost motion and torsional stiffness. Hu et al. [25] considered the ring flexibility of thin-walled gears, and divided the inner ring gear into multiple curved beams to establish the meshing stiffness model of a thin-walled flexible ring gear. In addition, he also analyzed the influence of the ring gear thickness and cross-sectional shape on meshing stiffness. Dong et al. [26] analyzed the elastic motion behavior of the flexible middle plane under no load and a small load. Subsequently, Gravagno et al. [27] proposed a new method to calculate the tension of the flexible neutral layer in HD and studied the relationship between the bending stress and circumferential strain of the flexible wheel and rollers of the wave generator. Tjahjowidodo [28] established a harmonic drive torsional compliance model to accurately capture the hysteresis in the torsional stiffness. Then, Zhang et al. [29] established a model for the compliance behavior of the flexible wheel to analyze the torsional compliance and stiffness of the harmonic drive system. Rheaume et al. [30,31] used finite element software to establish a numerical model of a harmonic gear to obtain the torsional stiffness and discussed the influence of geometric parameters on stiffness. Timofeev et al. [32] considered the deformation, processing error, and meshing characteristics of the flexible wheel to establish a mathematical model of harmonic gear transmission, and studied the torsional stiffness of HD. The study of this work on high-torque harmonic drives was beneficial to this paper. Ma et al. [33] established a FEM to study the meshing stiffness and torsional stiffness of HD. In addition, they also analyzed the influence of torque on the meshing teeth and meshing length. The study showed that the number of meshing teeth would increase with increasing load. Ma et al. [34] established an integrated system to analyze the meshing characteristics of a harmonic drive with multiple contacts among the teeth at the same time. Using a FEM, Wei et al. [35] combined the static and dynamic contact characteristics of the harmonic gear to obtain the meshing stiffness of HD. His work provided a basis for the dynamic analysis of HD. At present, research on the meshing stiffness of HD mostly adopts the finite element method, and the stiffness of the disposable harmonic gear is rarely involved. Therefore, it is necessary to propose a theoretical method to calculate the meshing stiffness of disposable harmonic gear.

Disposable machinery is a burgeoning development field. Different from the traditional HD, the disposable HD applied to a high load has a higher load capacity, smaller volume weight, and lower service life. In the previous study, we discussed the contact characteristics of the disposable HD from the point of strength and analyzed the no-load backlash, load distribution, and contact stress of HD [36]. However, research on the stiffness of the disposable HD has not been carried out. The combination of the research results on strength and stiffness can help to establish the design theory of disposable harmonic drive. Therefore, the comprehensive meshing stiffness of the disposable HD under full load operation will be studied in this work. Taking into account the cost of a disposable HD, the involute curve, which is more convenient to process, was selected as the flexible tooth profile in this work. In Section 2, the structure of the disposable harmonic gear is discussed. In Section 3, the non-uniform beam model and the improved energy method are used to calculate the single-tooth stiffness of a disposable harmonic drive under full load. Moreover, considering the influence of multi-tooth meshing, the improved energy method is modified to obtain the comprehensive meshing stiffness. Additionally, the stiffness curves obtained by the two methods are compared. Finally, a loaded harmonic gear is represented by a three-dimensional (3D) FEM, and the normal force and comprehensive elastic displacement while in engagement are extracted in Section 4. The failure mode of the disposable flexible wheel under overload is also discussed by FEM. The conclusions of this work are generalized in Section 5.

## 2. Design Scheme of the Disposable Harmonic Drive

### 2.1. Design of the Disposable Harmonic Flexible Wheel

The disposable HD consisted of a flexible wheel, a rigid wheel, and a wave generator. Figure 1 shows a conventional cup harmonic gear. A long cup flexible wheel was used as the conventional harmonic reducer to reduce the bottom stress concentration and extend the service life. However, this structure is contrary to the requirements of a high power–weight ratio for disposable harmonic gear. Considering that the disposable harmonic reducer is used for high loads and has a very low service life, and in order to improve the ultimate bearing capacity of a disposable harmonic drive with limited size, a straight flexible wheel with the complex wave transmission was selected (see Figure 2). This type of flexible wheel can be equivalent to a thin-walled external gear. It can increase the bearing torque of the disposable harmonic drive while compressing the axial dimension. In addition, it can also decrease the torsional hysteresis caused by the deformation of the cylinder to improve the transmission accuracy and meet the requirements of the disposable harmonic reducer. Therefore, the structure of a complex wave harmonic drive is more suitable for a disposable harmonic gear under full load operation.

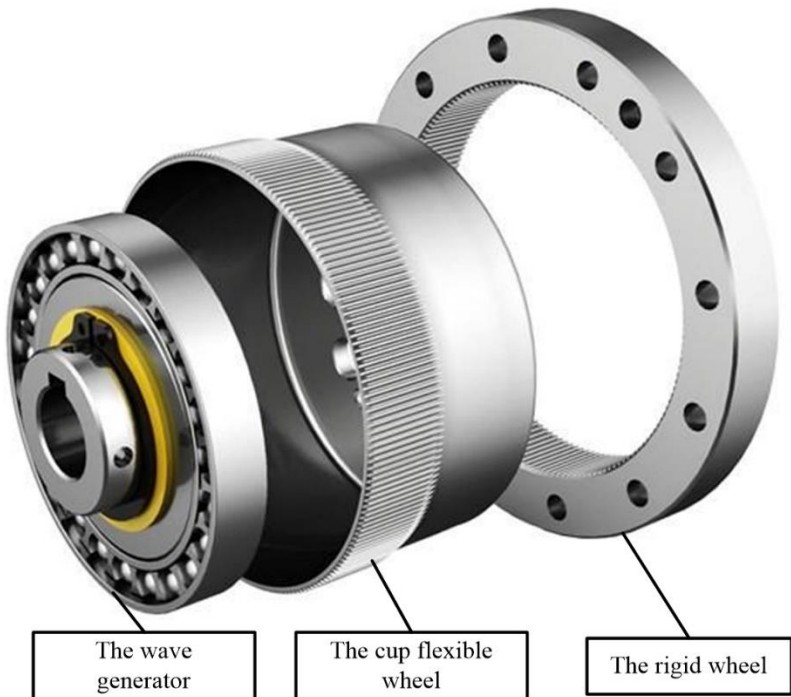

**Figure 1.** Conventional cup harmonic drive.

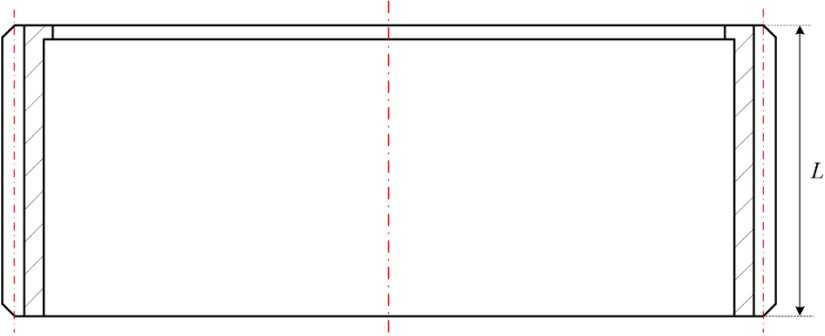

**Figure 2.** Structure of the disposable flexible wheel.

High-cycle fatigue failure is usually not considered in disposable HD, but low-cycle failure should be emphasized. Medium-carbon alloy steels, such as 40Cr, 40CrNiMo, and

30CrMnSiA, are the first choice for flexible wheels. Cr, Ni, Mo, Mn and Si can refine metal grains so as to improve the stiffness and toughness of steel. In this paper, 40CrNiMoA under a quenching–tempering process was selected as the disposable flexible wheel material. The stress–strain curve obtained by the tension and compression test of 40CrNiMoA is shown in Figure 3, and the yield strength $\sigma_s$ was determined as 960 MPa.

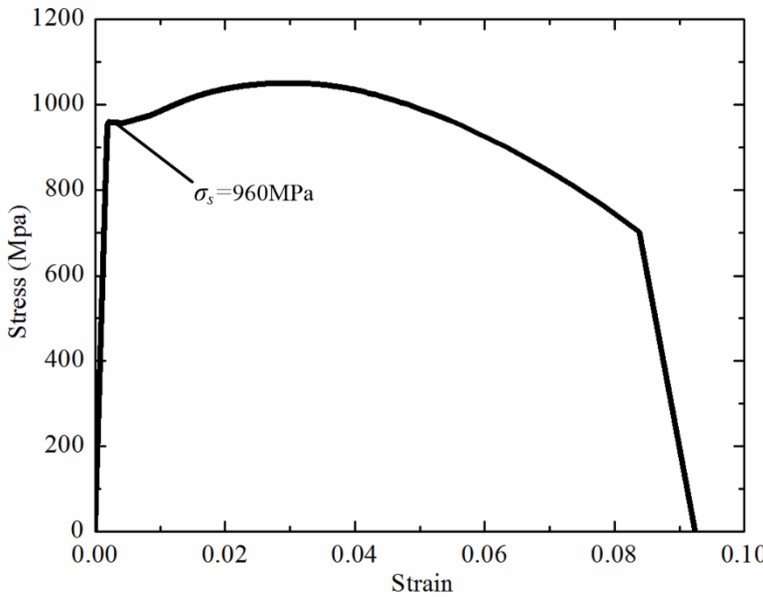

**Figure 3.** Tensile curve of 40CrNiMoA.

*2.2. Tooth Profile Design of the Disposable Harmonic Rigid Wheel*

According to the working characteristics of disposable harmonic drives, the following assumptions are made in this paper:

(1)  The distortion of the flexible wheel is not considered during transmission, and the length of the neutral layer of the flexible wheel remains unchanged;
(2)  The tooth profile of the flexible wheel does not change during assembly and transmission;
(3)  The symmetrical section of the flexible teeth is still perpendicular to the neutral layer of the flexible wheel after deformation;
(4)  The neutral layer of the flexible wheel remains stable during meshing.

Based on the above assumptions, on the premise that the flexible tooth profile and the shape of the wave generator have been determined, the envelope method is used to calculate the rigid tooth profile (see Figure 4). There are three coordinate systems in Figure 4: the wave generator coordinate system $C(O; x, y, z)$, the flexible wheel coordinate system $C_r(O_r; x_r, y_r, z_r)$, and the rigid wheel coordinate system $C_g(O_g; x_g, y_g, z_g)$. $O$ and $O_g$ coincide with the rotation center of the harmonic drive, while $O_r$ is the intersection of the symmetry line of the deformed flexible tooth and the neutral layer of the flexible wheel rim. $y_r$ is the symmetry axis of the flexible tooth and $y_g$ is the symmetry axis of the rigid tooth cogging. The original curve before harmonic gear assembly was set as $\widetilde{C}_r$ and the motion trajectory of a point on the original curve in $C_g$ was defined as $\widetilde{C}_r'$. Then, the envelope curve of the curve family when the flexible tooth profile $\widetilde{r}$ moved along $\widetilde{C}_r'$ in $C_g$ was the rigid wheel tooth profile.

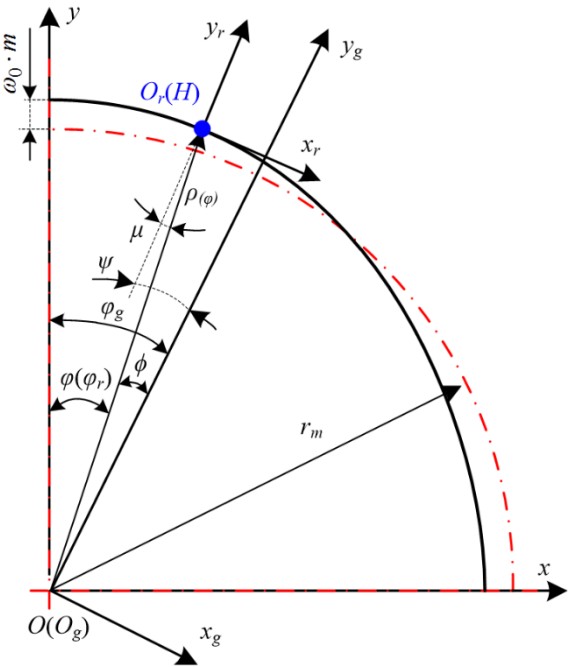

**Figure 4.** Coordinate system used in calculating the conjugate rigid tooth.

The elliptical cam wave generator is a wave generator with an ellipse as the basic profile of the cam. Compared with other wave generators, the elliptical cam wave generator can ensure the excellent performance of the harmonic drive and is easy to process. The function of the original curve after assembly can be expressed as:

$$\rho_{(\varphi)} = \sqrt{\left(r_m + \omega_0^* m\right)^2 - 4 r_m \omega_0^* m \sin^2 \varphi} \tag{1}$$

where $r_m$ represents the neutral layer radius of the flexible wheel rim, $\omega_0^*$ denotes the radial deformation coefficient of the flexible wheel, and $m$ is the gear module.

The radial deformation of point $H$ at the flexible wheel rim neutral layer after assembly can be expressed as:

$$\omega_{(\varphi)} = \sqrt{\left(r_m + \omega_0^* m\right)^2 - 4 r_m \omega_0^* m \sin^2 \varphi} - r_m \tag{2}$$

According to Equation (1), the included angle between the vector radius and the curvature radius traversing through point $H$ can be expressed as:

$$\mu_{(\varphi)} = \arctan \frac{\rho'}{\rho_{(\varphi)}} = \arctan \frac{\rho'}{r_m + \omega_{(\varphi)}} = \frac{1}{r_m} \frac{d\omega_{(\varphi)}}{d\varphi} \tag{3}$$

In the flexible coordinate system, the flexible tooth profile curve can be calculated with:

$$\begin{cases} x_r = x_r(\mu) \\ y_r = y_r(\mu) \end{cases} \tag{4}$$

The flexible tooth profile and wave generator profile curve are shown in Figures 5 and 6, respectively.

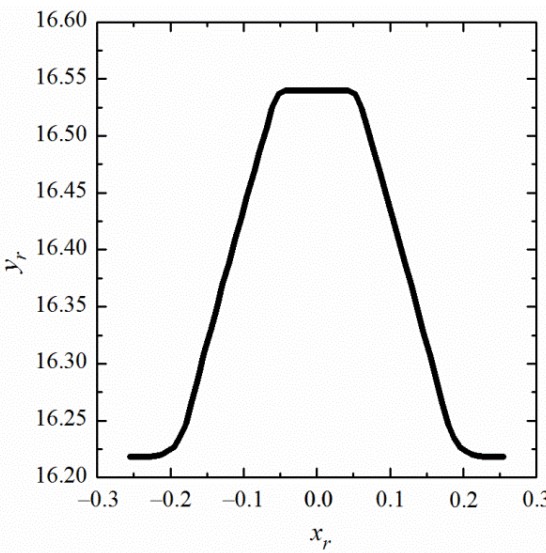

**Figure 5.** Tooth profile of the disposable flexible wheel.

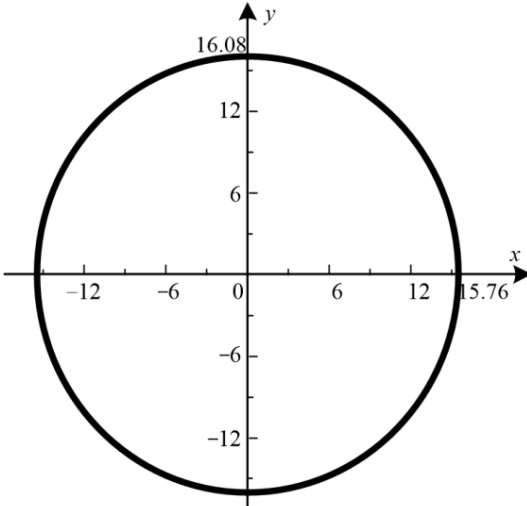

**Figure 6.** Profile curve of the elliptical cam wave generator.

According to the envelope theory of harmonic drive, the rigid tooth profile conjugated with the disposable flexible wheel can be obtained as:

$$
\begin{cases}
x_g(\mu, \varphi) = x_r(\mu) \cos \psi + y_r(\mu) \sin \psi + \rho_{(\varphi)} \sin \gamma \\
y_g(\mu, \varphi) = -x_r(\mu) \sin \psi + y_r(\mu) \cos \psi + \rho_{(\varphi)} \cos \gamma \\
\frac{\partial x_g(\mu,\varphi)}{\partial \mu} \cdot \frac{\partial y_g(\mu,\varphi)}{\partial \varphi} - \frac{\partial x_g(\mu,\varphi)}{\partial \varphi} \cdot \frac{\partial y_g(\mu,\varphi)}{\partial \mu} = 0 \\
\psi_{(\varphi)} = \mu + \phi \\
\phi_{(\varphi)} = \varphi_r - \varphi_g
\end{cases}
\tag{5}
$$

For the disposable harmonic drive, $\varphi_r$ and $\varphi_g$ can be expressed as:

$$
\begin{cases}
\varphi_r = \varphi \\
\varphi_g = \frac{Z_r}{Z_g} \cdot \varphi
\end{cases}
\tag{6}
$$

where $Z_r$ and $Z_g$ indicate the tooth number of the flexible and rigid wheels, respectively.

Then, the transformation matrix from the flexible wheel coordinate system to the rigid wheel coordinate system can be expressed as:

$$M_{rg} = \begin{bmatrix} \cos\psi & \sin\psi & \rho\sin\gamma \\ -\sin\psi & \cos\psi & \rho\cos\gamma \\ 0 & 0 & 1 \end{bmatrix} \tag{7}$$

The discretization of the flexible tooth profile curve was substituted into Equation (8), and a series of curve clusters of $0 \le \varphi \le \pi/2$ were obtained. Then, the envelope curve of the curve cluster was obtained through program calculation, which was defined as the rigid tooth profile.

$$\begin{bmatrix} x_g \\ y_g \\ 1 \end{bmatrix} = M_{rg} \begin{bmatrix} x_r \\ y_r \\ 1 \end{bmatrix} \tag{8}$$

The rigid tooth profile obtained by the envelope method is shown in Figure 7. The thin curve family in this figure is the motion trajectory of the flexible tooth. And the solid blue line is the tooth profile of the rigid wheel after enveloping.

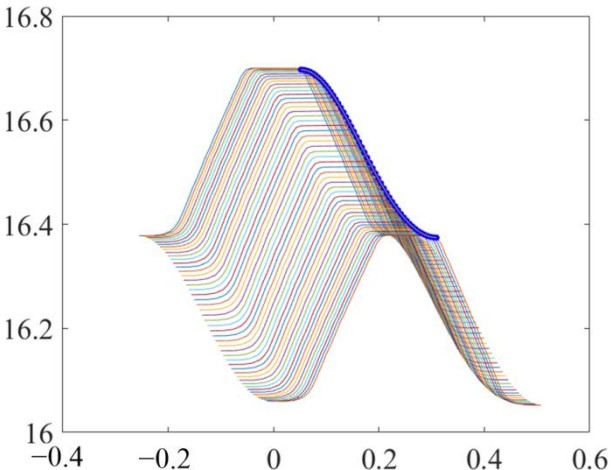

**Figure 7.** Rigid tooth profile fitted by the envelope method.

After data-fitting, the rigid tooth profile curve of the disposable harmonic drive in this paper can be expressed as:

$$y_g(x) = 18.732x^3 - 7.995x^2 - 1.2504x + 16.82 \tag{9}$$

### 3. Analytical Model to Compute the Meshing Stiffness of the Disposable Harmonic Drive under Full Load

The flexible wheel in the harmonic drive is a thin-walled component. The thickness of the rim is similar to the height of each of its teeth. According to Refs. [16,24], the thin rim has an effect on meshing stiffness. In addition, the simultaneous meshing of multiple teeth during the disposable harmonic drive will also affect the stiffness. Thus, prevalent methods cannot accurately calculate the stiffness of the flexible wheel. Because the rim of the rigid wheel is much thicker than the tooth, the stiffness of the rigid wheel can be calculated by the potential energy method for a cylindrical gear.

*3.1. Stiffness of the Flexible Wheel Tooth*

The single-tooth stiffness of gear transmission can be expressed by the elastic deformation of a single gear tooth in the meshing process. For the disposable harmonic gear, the elastic deformation mainly includes tooth root bending deformation, shear deformation, and tooth surface contact deformation.

The general expression for the stiffness of single-tooth transmission is as follows:

$$k = \frac{F}{\delta} \tag{10}$$

where $F$ is the transmission force acting on the tooth and $\delta$ denotes the comprehensive displacement along the direction of the force.

The equivalent model of the flexible wheel is shown in Figure 8. In this figure, *AM* and *BN* are the curves of the involute profile. The single-tooth model of the involute profile is shown in Figure 9. Along the action line, $F$ can be decomposed into axial component $F_{a1}$ and radial component $F_{b1}$. Compression energy is generated under the action of $F_{a1}$, shear energy is generated under the action of $F_{b1}$, and bending energy is generated by a combination of $F_{b1}$ and the additional bending moment $M_{x1}$. $F_{a1}$, $F_{b1}$, and $M_{x1}$ can be calculated as:

$$F_{a_1} = F \cdot \sin \theta_d \tag{11}$$

$$F_{b_1} = F \cdot \cos \theta_d \tag{12}$$

$$M_{x_1} = F_{b_1} \cdot (h_\delta + d - x) - F_{a_1} \cdot S_F \tag{13}$$

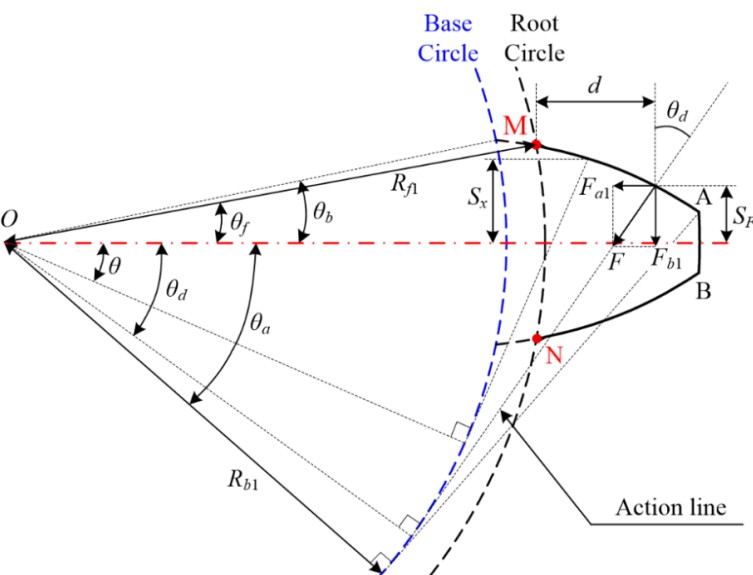

**Figure 8.** Equivalent model of the involute flexible wheel.

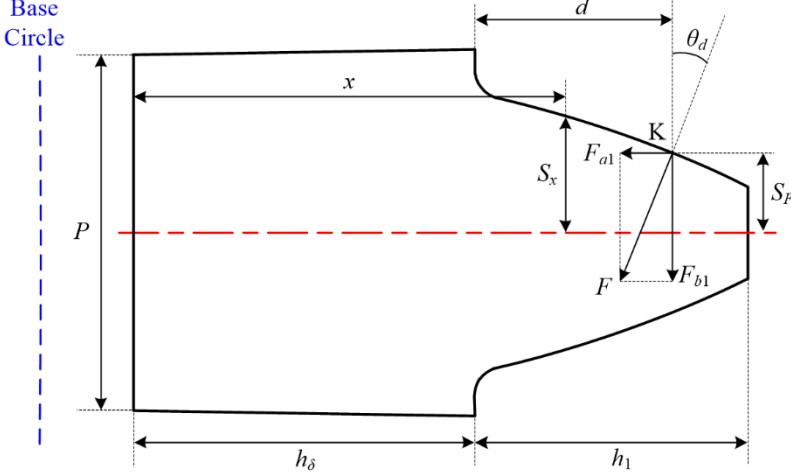

**Figure 9.** Single-tooth beam model of the involute flexible wheel.

According to beam theory and the Cartesian theorem, the energy stored in a single flexible tooth can be expressed as follows:

$$U_\varepsilon = \int_0^{h_\delta} \frac{F_{a_1}^2}{2EA_{1p}}dx + \int_0^{h_\delta} \frac{F_{b_1}^2}{2GA_{1p}}dx + \int_0^{h_\delta} \frac{M_{x_1}^2}{2EI_{1p}}dx + \int_{h_\delta}^{h_\delta+d} \frac{F_{a_1}^2}{2EA_{1x}}dx + \int_{h_\delta}^{h_\delta+d} \frac{F_{b_1}^2}{2GA_{1x}}dx + \int_{h_\delta}^{h_\delta+d} \frac{M_{x_1}^2}{2EI_{1x}}dx \quad (14)$$

where $E$ and $G$ represent Young's modulus and the shear modulus, respectively; $h_1$ denotes the height of the tooth of the gear; $h_\delta$ describes the thickness of the rim of a flexible wheel; $A_{1x}$ and $I_{1x}$ are the area and the area moment of inertia of the section where the distance to the bottom of the rim is $x$, respectively; and $A_{1p}$ and $I_{1p}$ describe the area and the area moment of inertia of the rim section of a flexible wheel, respectively (see Figure 8).

To simplify the calculation, $A_{1x}$, $I_{1x}$, $A_{1p}$ and $I_{1p}$ can be expressed as follows:

$$A_{1x} = 2S_x L \quad (15)$$

$$I_{1x} = \frac{1}{12}(2S_x)^3 L \quad (16)$$

$$A_{1p} = PL \quad (17)$$

$$I_{1p} = \frac{1}{12}P^3 L \quad (18)$$

where $S_x$ denotes half of the thickness of a given flexible tooth where the distance to the bottom of the rim is $x$, $S_F$ is half of the tooth thickness at the action position $K$, $P$ is the pitch of a flexible tooth, and $L$ represents its width.

Substituting Equations (12)–(14) and (16)–(19) into Equation (15), the comprehensive displacement of a single flexible tooth can be expressed as follows:

$$\delta_\varepsilon = \frac{F\sin^2\theta_d}{EPL}h_\delta + \frac{F\cos^2\theta_d}{GPL}h_\delta + \frac{F}{\frac{1}{12}EP^3L}\int_0^{h_\delta}[(h_\delta+d-x)\cos\theta_d - S_F\sin\theta_d]^2 dx + \\ \frac{F\sin^2\theta_d}{EL}\int_{h_\delta}^{h_\delta+d}\frac{dx}{S_x} + \frac{F\cos^2\theta_d}{GL}\int_{h_\delta}^{h_\delta+d}\frac{dx}{S_x} + \frac{F}{\frac{2}{3}EL}\int_{h_\delta}^{h_\delta+d}\frac{[(h_\delta+d-x)\cos\alpha_d - S_F\sin\theta_d]^2}{S_x^3}dx \quad (19)$$

According to the characteristics of the involute curve, the angular position variable $\theta$ is introduced, and $S_x$ and $S_F$ can then be expressed as follows:

$$\begin{aligned} S_x &= R_{b_1}[(\theta_b - \theta)\cos\theta + \sin\theta], \\ S_F &= R_{b_1}[(\theta_b + \theta_d)\cos\theta - \sin\theta_d] \end{aligned} \quad (20)$$

where $\theta_b$ describes the half tooth angle on the base circle:

$$\theta_b = \frac{\pi}{2Z_1} + inv\alpha_0 \quad (21)$$

where $\alpha_0$ denotes the pressure angle, and $Z_1$ is the tooth number of the flexible wheel.

By substituting Equations (19) and (20) into Equation (10), the stiffness of a single flexible tooth of the involute profile can be expressed as follows:

$$k_r = \frac{F}{\delta_\varepsilon} = \frac{EL}{C_{i_1}\sin^2\theta_d + C_{i_2}\cos^2\theta_d + C_{i_3}\sin\theta_d\cos\theta_d} \quad (22)$$

where $C_{i1}$, $C_{i2}$, and $C_{i3}$ denote the relevant parameters of the tooth and can be expressed as follows:

$$C_{i_1} = C_1 + C_2 \cdot C_{a_1} + \frac{5}{2}C_{a_2} \quad (23)$$

$$C_{i_2} = 2(1+v)C_1 + C_2 \cdot C_{b_1} + 2(1+v) \cdot C_{ab} + C_{b_2} \quad (24)$$

$$C_{i_3} = -(2C_2 \cdot C_{s_1} + 3C_{s_2}) \quad (25)$$

$$C_1 = \frac{h_\delta}{P}, \; C_2 = \frac{12}{P3} \tag{26}$$

$$C_{a_1} = \int_{\theta_f}^{\theta_\delta} R_{b_1}^3 [(\theta_b - \theta)cos\theta + \sin\theta]^2 (\theta - \theta_b)cos\theta d\theta,$$
$$C_{ab} = \int_{\theta_f}^{\theta_\delta} \frac{(\theta - \theta_b)cos\theta}{(\theta_b - \theta)cos\theta + \sin\theta} d\theta \tag{27}$$

$$C_{b_1} = \int_{\theta_f}^{\theta_\delta} R_{b_1}^3 [\cos\theta_d - \cos\theta + (\theta_b - \theta)\sin\theta - (\theta_b - \theta_d)\sin\theta_d]^2 (\theta - \theta_b)cos\theta d\theta,$$
$$C_{b_2} = \int_{-\theta_d}^{\theta_f} \frac{[\cos\theta_d - \cos\theta + (\theta_b - \theta)\sin\theta - (\theta_b - \theta_d)\sin\theta_d]^2 (\theta - \theta_b)cos\theta}{[(\theta_b - \theta)cos\theta + sin\theta]^3} d\theta \tag{28}$$

$$C_{s_1} = \int_{\theta_f}^{\theta_\delta} R_{b_1}^3 [\cos\theta_d - \cos\theta + (\theta_b - \theta)\sin\theta - (\theta_b - \theta_d)\sin\theta_d][(\theta_b - \theta)cos\theta + \sin\theta](\theta - \theta_b)cos\theta d\theta,$$
$$C_{s_2} = \int_{-\theta_d}^{\theta_f} \frac{[\cos\theta_d - \cos\theta + (\theta_b - \theta)\sin\theta - (\theta_b - \theta_d)\sin\theta_d](\theta - \theta_b)cos\theta}{[(\theta_b - \theta)cos\theta + sin\theta]^2} d\theta, \tag{29}$$

### 3.2. Stiffness of the Rigid Wheel Tooth

The single-tooth model of the rigid wheel is shown in Figure 10. Different from the involute tooth profile of the flexible wheel, the tooth profile of the rigid wheel is obtained according to the envelope method (see Equation (9)).

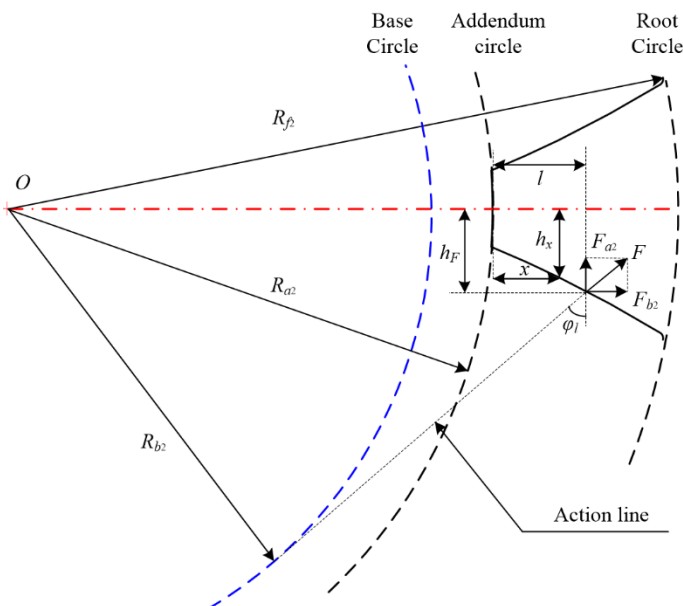

**Figure 10.** Equivalent model of the involute rigid wheel.

The orthogonal component of the action force $F$ and the equivalent bending moment can be expressed as:

$$F_{a_2} = F \cdot \sin\varphi_l \tag{30}$$

$$F_{b_2} = F \cdot \cos\varphi_l \tag{31}$$

$$M_{x_2} = F_{b_2} \cdot (l - x) - F_{a_2} \cdot h_F \tag{32}$$

By applying the beam theory, the bending, axial compressive, and shear energies stored in a rigid tooth can be obtained as follows:

$$U_{b_2} = \frac{F^2}{2k_{b_2}} = \int_0^l \frac{M_{x_2}^2}{2EI_{2x}} dx \tag{33}$$

$$U_{a_2} = \frac{F^2}{2k_{a_2}} = \int_0^l \frac{F_{a_2}^2}{2EA_{2x}} dx \tag{34}$$

$$U_{s_2} = \frac{F^2}{2k_{s_2}} = \int_0^l \frac{1.2F_{b_2}{}^2}{2GA_{2x}}dx \tag{35}$$

where $A_{2x}$ and $I_{2x}$ are the area and the area moment of inertia of the section where the distance to the top of the rigid tooth is $x$, respectively. They can be expressed as follows:

$$A_{2x} = 2h_x L \tag{36}$$

$$I_{2x} = \frac{1}{12}(2h_x)^3 L \tag{37}$$

where $h_x$ denotes half of the thickness of the rigid tooth where the distance to the tooth top is $x$. Therefore, the bending stiffness ($k_{b2}$), axial compressive stiffness ($k_{a2}$), and shear stiffness ($k_{s2}$) of the rigid tooth are given as:

$$\frac{1}{k_{b_2}} = \int_0^l \frac{[\cos \varphi_l \cdot (l-x) - \sin \varphi_l \cdot h_F]^2}{\frac{2}{3}ELh_x^3}dx \tag{38}$$

$$\frac{1}{k_{b_2}} = \int_0^l \frac{[\cos \varphi_l \cdot (l-x) - \sin \varphi_l \cdot h_F]^2}{\frac{2}{3}ELh_x^3}dx \tag{39}$$

$$\frac{1}{k_{s_2}} = \int_0^l \frac{1.2\cos^2 \varphi_l \cdot 2(1+v)}{2ELh_x}dx \tag{40}$$

where $h_F$ denotes half of the thickness of the rigid tooth where the distance to the tooth top is $l$. According to the fitting curve of the rigid wheel obtained from Equation (9) in Section 2.2, $h_x$ and $h_F$ can be expressed as follows:

$$h_x = -0.5378(R_{a2} - x)^3 + 26.666(R_{a2} - x)^2 - 441.16(R_{a2} - x) + 2435.3 \tag{41}$$

$$h_F = -0.5378(R_{a2} - l)^3 + 26.666(R_{a2} - l)^2 - 441.16(R_{a2} - l) + 2435.3 \tag{42}$$

The fillet foundation displacement in the direction of tooth load can be obtained by Sainsot et al. [8] as:

$$\delta_{f_2} = \frac{F\cos^2 \varphi_l}{EL} \cdot \left\{ L^* \left( \frac{U_f}{S_f} \right)^2 + M^* \left( \frac{U_f}{S_f} \right) + P^* \left( 1 + Q^* \tan^2 \varphi_l \right) \right\} \tag{43}$$

where $U_f$ and $S_f$ are as shown in Figure 11, and $L^*$, $M^*$, $P^*$, and $Q^*$ are constants that differ slightly depending on the assumptions shown in Table 1.

**Table 1.** Values of the coefficients of Equation (43).

|  | L* | M* | P* | Q* |
|---|---|---|---|---|
| Weber [5]–Attia [6] | 5.2 | 1 | 1.4 | 0.294–0.32 |
| Cornell [7] | 5.306 | 1.4 (plane stress) 1.14 (plane strain) | 1.534 | 0.32 |

The stiffness considering the gear fillet foundation deflection can be expressed as:

$$\frac{1}{k_{f_2}} = \frac{\delta_{f_2}}{F} = \frac{\cos^2 \varphi_l}{EL} \cdot \left\{ L^* \left( \frac{U_f}{S_f} \right)^2 + M^* \left( \frac{U_f}{S_f} \right) + P^* \left( 1 + Q^* \tan^2 \varphi_l \right) \right\} \tag{44}$$

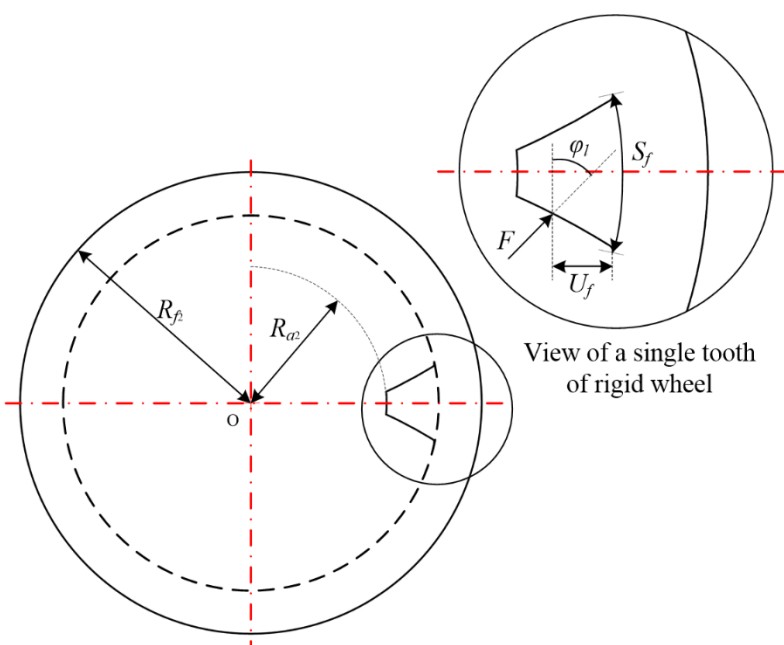

**Figure 11.** Geometrical parameters of $k_{f2}$.

The single-tooth stiffness of the rigid wheel can be obtained as:

$$k_g = \frac{1}{\frac{1}{k_{b_2}} + \frac{1}{k_{a_2}} + \frac{1}{k_{s_2}} + \frac{1}{k_{f_2}}} \tag{45}$$

The parameters in Table 2 were used to model the harmonic gears. The equivalent meshing stiffness of a tooth pair in transmission can be calculated by:

$$\frac{1}{k} = \frac{1}{k_r} + \frac{1}{k_g} + \frac{1}{k_h} \tag{46}$$

**Table 2.** Parameters of the HD.

| Parameter | Flexible Wheel | Rigid Wheel |
|---|---|---|
| Number of teeth $Z$ | 200 | 202 |
| Module $m$ (mm) | 0.16 | 0.16 |
| Teeth width $L$ (mm) | 10 | 10 |
| Pressure angle $\alpha_0$ (°) | 20 | |
| Transmission ratio | 100 | |

The Hertz contact stiffness $k_h$ is given by Yang et al. [3] as:

$$k_h = \frac{\pi E L}{4(1 - \nu^2)} \tag{47}$$

where $\nu$ describes the Poisson's ratio of the material of a rigid wheel.

The above-mentioned procedure can be repeated when multiple pairs of teeth are in contact. The comprehensive stiffness can be expressed as:

$$K = k_1 + k_2 + \cdots k_n \tag{48}$$

where $n$ denotes the contact teeth number in the meshing region.

The single-tooth stiffness of the disposable harmonic drive in the involute profile as obtained by the improved energy method is shown in Figure 12. The stiffness obtained by the improved energy method, proposed in this article, was compared with the stiffness

calculated by the analytical method without considering the thin rim in Ref. [10], as shown in Figure 13.

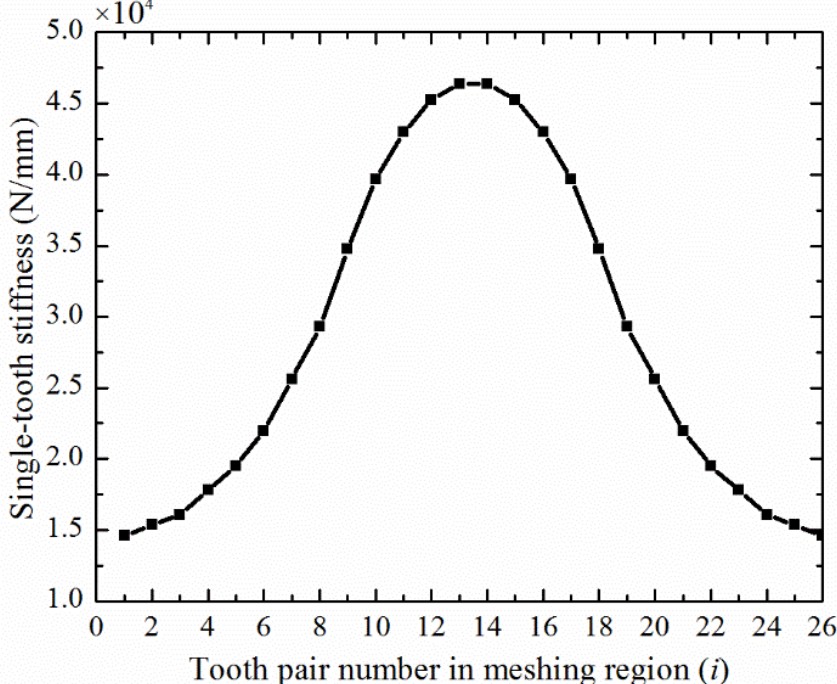

**Figure 12.** Single-tooth stiffness of the disposable harmonic gear.

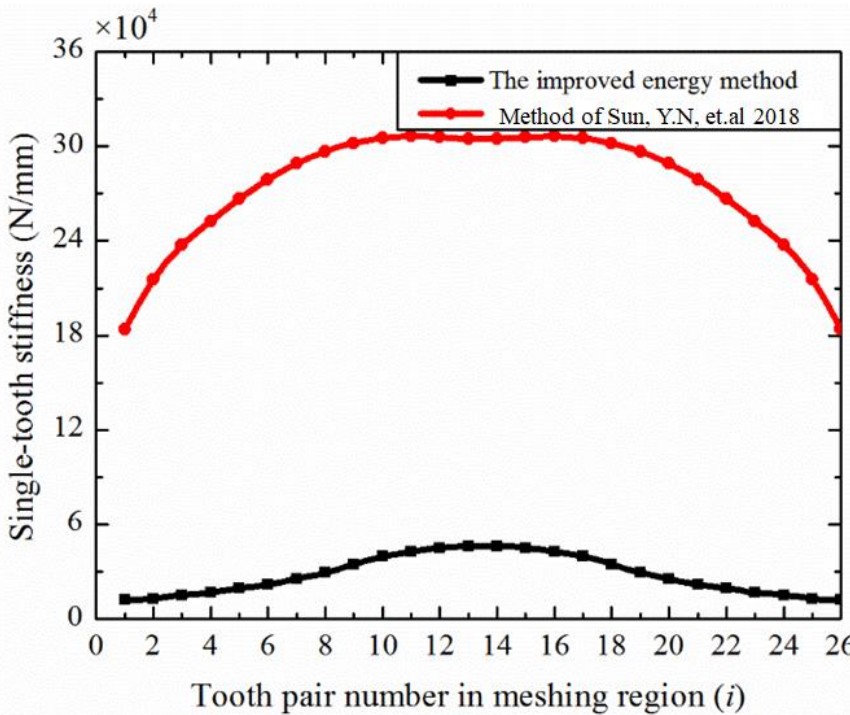

**Figure 13.** Single-tooth stiffness obtained by two analytical methods [10].

Compared with conventional spur gears, the extremely thin rim structure of disposable harmonic gears will have a great influence on their stiffness, which must be considered. In addition, the proportion of the teeth number involved in the meshing of the disposable harmonic HD can be close to 30%. Therefore, in order to obtain the comprehensive stiffness of the disposable HD, the influence of the remaining teeth that are meshed at the same time in a meshing cycle should also be considered.

### 3.3. Stiffness of Multi-Tooth Meshing

Under the action of the elliptical cam wave generator, the rim of the flexible wheel is stretched from a circle to an ellipse. Therefore, the flexible wheel is divided into a contact area and a non-contact area, and $\gamma$ represents the range of the contact area (see Figure 14a). The micro-unit on the flexible rim at position $\varphi$ in the contact area is acted on by the radial force $q_r$ generated by the wave generator. The schematic diagram of internal force calculation is shown in Figure 14b.

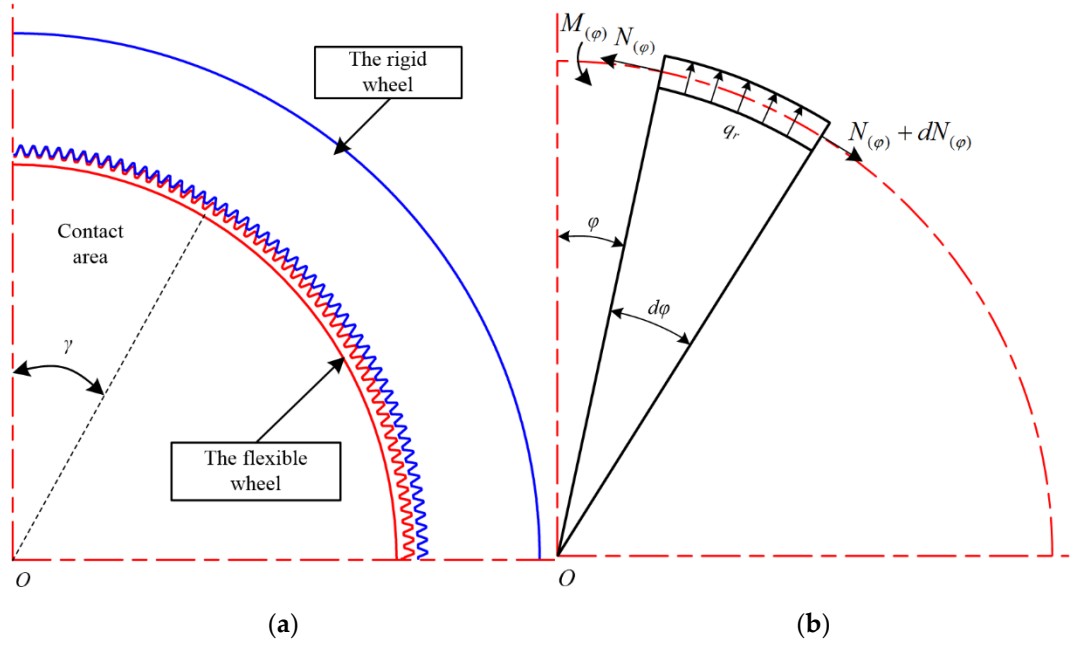

**Figure 14.** (**a**) Contact region distribution of the disposable HD; (**b**) internal force calculation schematic diagram of the flexible rim in the contact area.

When the disposable harmonic gear is loaded, the transmission torque $T$ acting on the flexible wheel can be expressed as:

$$T = 4 \int_0^\gamma \left( \frac{d_1}{2} \right)^2 L q_{t\max} \cos \left( \frac{\pi \varphi}{2\gamma} \right) d\varphi \tag{49}$$

where $q_t$ is the circumferentially distributed load per unit width of the flexible wheel rim, which can be expressed as:

$$q_{t\max} = \frac{\pi T}{2\gamma d_1^2 L} \tag{50}$$

where $d_1$ represents the diameter of the flexible wheel index circle.

According to the equilibrium equation:

$$q_{t\max} = \frac{\pi T}{2\gamma d_1^2 L} \tag{51}$$

where $q_r = q_{t\max} \cdot q_r^*$, and $q_r^*$ indicates a dimensionless coefficient, which can be expressed as $q_r^* = 0.375[1 - \sin(\varphi/2)]$

Therefore, the tension of the flexible rim in the contact area can be expressed as:

$$N_{(\varphi)} = q_{t\max} \cdot q_r^* \cdot r_m \ (0 \le \varphi \le \gamma) \tag{52}$$

The tensile stress can be expressed as:

$$\sigma_t = \frac{N_{(\varphi)}}{h_\delta} = \frac{0.375\left(1 - \sin\frac{\varphi}{2}\right)\pi T r_m}{2\gamma d_1^2 L h_\delta} \tag{53}$$

At the $i$-th pair of meshing teeth, the circumferential displacement of the flexible rim caused by tension can be expressed as follows:

$$\nu_t = \int_{\varphi_{i-1}}^{\varphi_i} r_m \cdot \frac{\sigma_t}{E} d\varphi = \frac{0.375\pi T r_m^2}{2E\gamma d_1^2 L h_\delta}\left[\left(\varphi_i + 2\cos\frac{\varphi_i}{2}\right) - \left(\varphi_{i-1} + 2\cos\frac{\varphi_{i-1}}{2}\right)\right] \tag{54}$$

According to the relationship between the bending moment and the curvature variable, the bending moment of the rim unit at the position $\varphi$ of the contact area can be expressed as:

$$M_{(\varphi)} = EI_z\left(\frac{1}{\rho_{(\varphi)}} - \frac{1}{r_m}\right) (0 \leq \varphi \leq \gamma) \tag{55}$$

where $EI_z$ is the circumferential bending stiffness of the flexible rim. The bending equation of the flexible rim in the contact area can be expressed as:

$$\frac{d^2\omega'}{d\varphi^2} + \omega'_{(\varphi)} = \frac{-M_{(\varphi)}r_m^2}{EI_z} (0 \leq \varphi \leq \gamma) \tag{56}$$

Substituting Equation (55) into Equation (56):

$$\omega'_{(\varphi)} = A\sin\varphi + B\cos\varphi - r_m^2\left(\frac{1}{\rho_{(\varphi)}} - \frac{1}{r_m}\right) (0 \leq \varphi \leq \gamma) \tag{57}$$

According to the boundary conditions:

$$\begin{cases} \omega'_{(0)} = \omega_0^* m \\ \frac{d\omega'}{d\varphi}\Big|_{\varphi=0} = 0 \end{cases} \tag{58}$$

Parameters $A$ and $B$ can be expressed as:

$$\begin{cases} A = 0 \\ B = \omega_0^* m + r_m^2\left(\frac{1}{\rho_{(\varphi)}} - \frac{1}{r_m}\right) \end{cases} \tag{59}$$

According to the non-elongation condition of the neutral layer,

$$\nu'_{(\varphi)} = \int_{\varphi_{i-1}}^{]\varphi_i} \omega'_{(\varphi)} d\varphi = -\frac{(M\sin\varphi_i - N\varphi_i) - (M\sin]\varphi_{i-1} - N]\varphi_{i-1})}{M - N} (0 \leq \varphi \leq \gamma) \tag{60}$$

$$M = -\sin\gamma\cos\gamma, \ N = \pi/4[\cos\gamma - (\pi/2 - \gamma)\sin\gamma] \tag{61}$$

Combining Equation (54) with Equation (60), the additional tangential displacement at the $i$-th pair of meshing teeth of the flexible wheel in the contact area caused by multi-tooth meshing under a load can be expressed as (see Figure 15):

$$\nu_{(i)} = -\frac{(M\sin\varphi_i - N\varphi_i) - (M\sin\varphi_{i-1} - N\varphi_{i-1})}{M - N} - \frac{0.375\pi T r_m^2}{2E\gamma d_1^2 L h_\delta}\left[\left(\varphi_i + 2\cos\frac{\varphi_i}{2}\right) - \left(\varphi_{i-1} + 2\cos\frac{\varphi_{i-1}}{2}\right)\right] \tag{62}$$

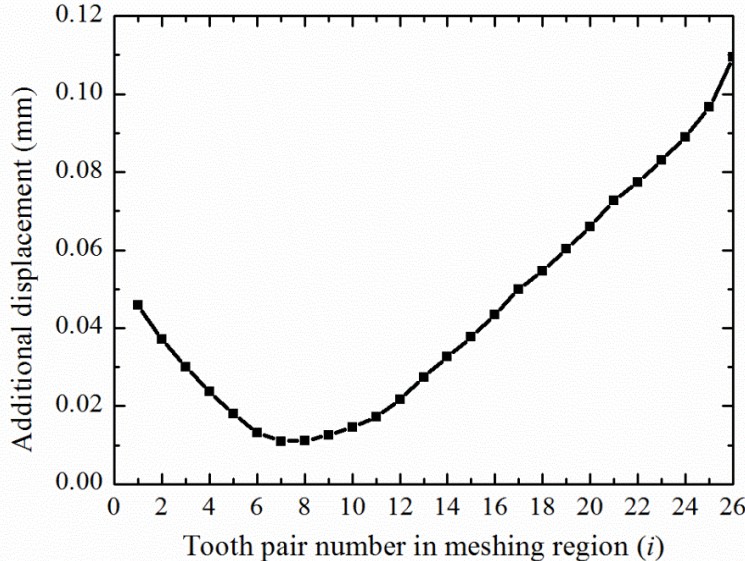

**Figure 15.** Additional tangential displacement of flexible wheel teeth in the meshing region.

With the continuous meshing of subsequent gear teeth, the deformation of flexible teeth in the meshing region decreases slowly and then increases gradually. After considering the additional displacement, the modified disposable harmonic gear meshing stiffness is shown in Figure 16.

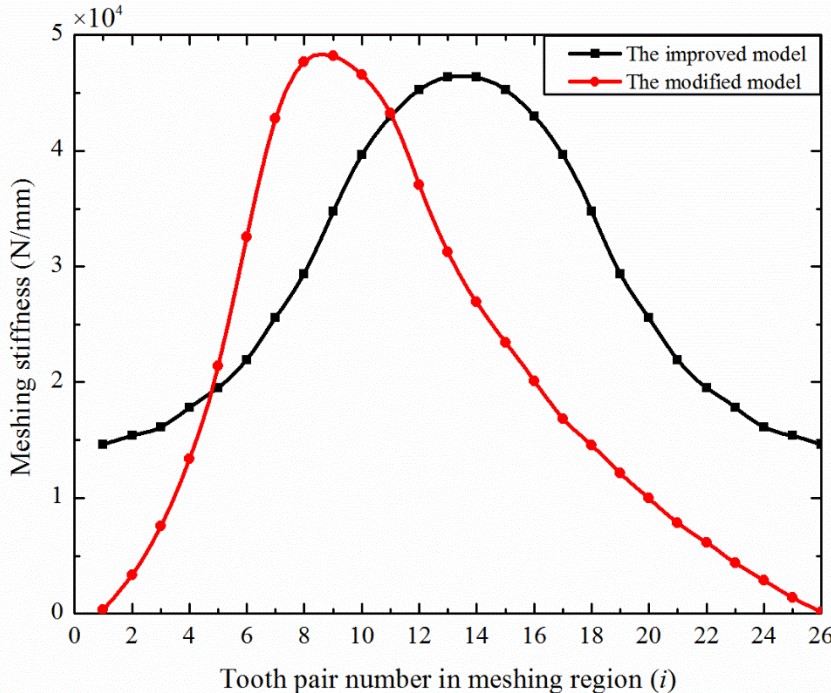

**Figure 16.** Meshing stiffness of the disposable HD obtained by two methods.

As can be seen from Figure 6, the two curves were similar in amplitudes. However, the modified meshing stiffness curve was not symmetrical, and the tooth position with the largest stiffness in the meshing region was on the left side of the center line. This was due to the additional deformation of the flexible wheel teeth caused by the multi-tooth meshing of the disposable harmonic gear.

## 4. Meshing Stiffness Using Finite Element Model

The harmonic gear pair considered here contained two types of gears: a flexible wheel and a rigid wheel. In addition, the wave generator that caused the periodic deformation of the flexible wheels was also included in the disposable HD. The harmonic gears were modeled in 3D simulation software. The corresponding performance parameters of the three parts are shown in Table 3.

**Table 3.** Parameters of the flexible wheel, rigid wheel and wave generator.

|  | Flexible Wheel | Rigid Wheel and Wave Generator |
|---|---|---|
| Material | 40CrNiMoA | 45 Steel |
| Density $\rho$ (kg/m$^3$) | 7850 | 7870 |
| Young's modulus $E$ (MPa) | 211,000 | 209,000 |
| Poisson's ratio $\nu$ | 0.3 | 0.27 |

The simplified model for the finite elements of the disposable HD is shown in Figure 17. The finite element analysis included two stages of assembly and loading. In order to apply boundary conditions and loads, reference points were set up at the center positions of the three components. Then, coupling constraints on the flexible internal surface, the external surface of the rigid wheel, and the wave generator with the corresponding reference points were established. The wave generator was treated as completely rigid during simulation. The FEM contained two types of contact: contact between the flexible internal surface and the external surface of the wave generator, and contact between the tooth surfaces of the two gears. The internal surface and the tooth surface of the flexible wheel were set as the slave surface, and the wave generator external surface and the rigid tooth surface were set as the master surface.

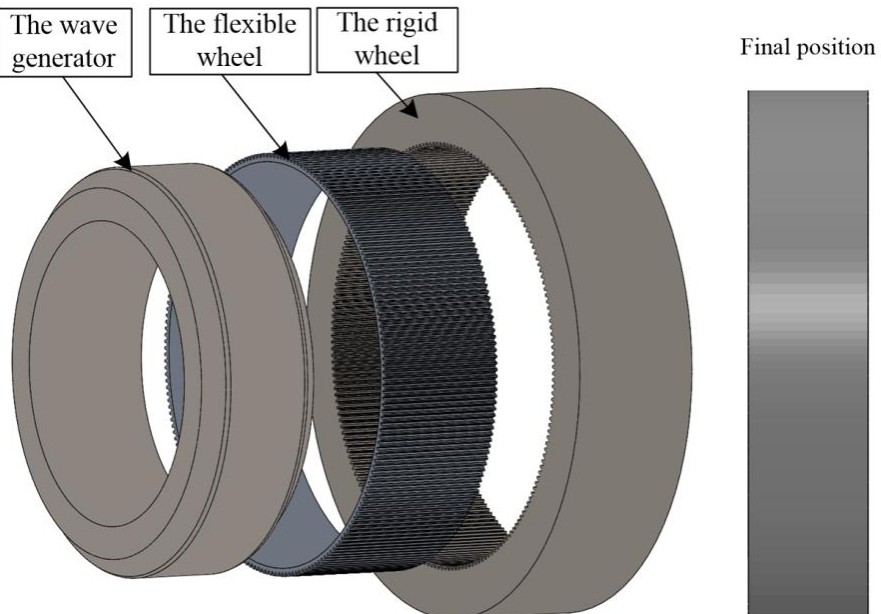

**Figure 17.** Simplified model of the disposable harmonic drive.

In step assembly, fix the flexible wheel and then move the other two components to the position matched with the flexible wheel at a uniform translation speed. In step loading, fix the rigid wheel's external surface. Then, apply a constant rotation speed to the other two components. Additionally, apply a full load of 80 N·m to the flexible wheel.

The FEM after the assembly of the disposable HD is shown in Figure 18. Figure 18a shows the position of the three components after assembly, and Figure 18b shows the magnification of several meshing tooth pairs in the contact area. The surfaces of the rigid

and flexible teeth are defined as the master surface and slave surfaces, respectively. The equivalent stress and deformation of the flexible wheel are shown in Figure 19.

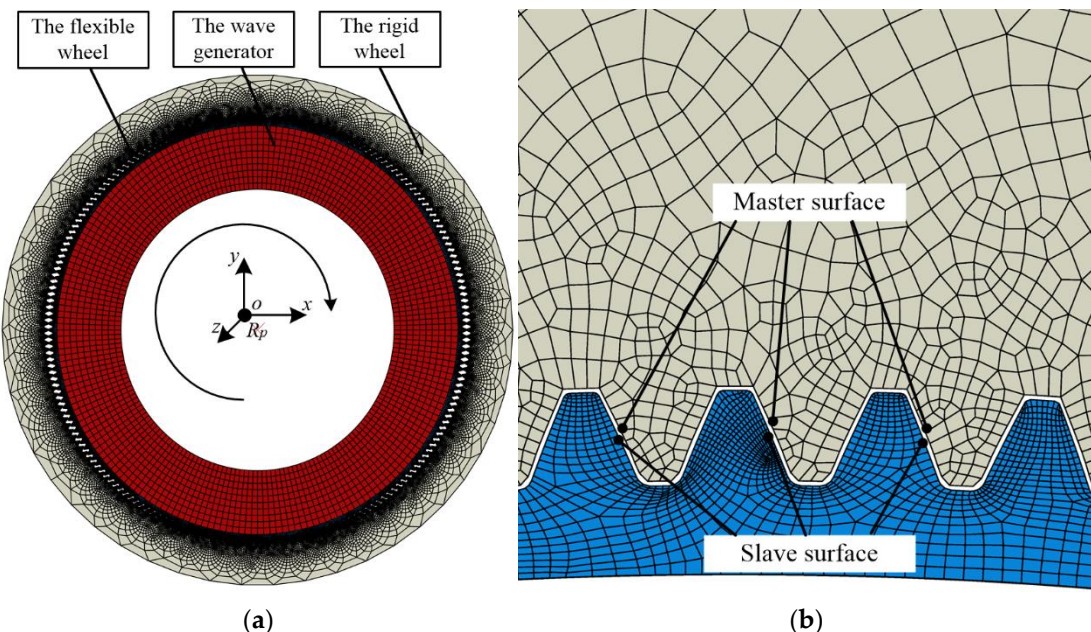

**Figure 18.** (**a**) Finite element assembly model of the disposable HD; (**b**) partial enlargement of the meshing region between the two gears.

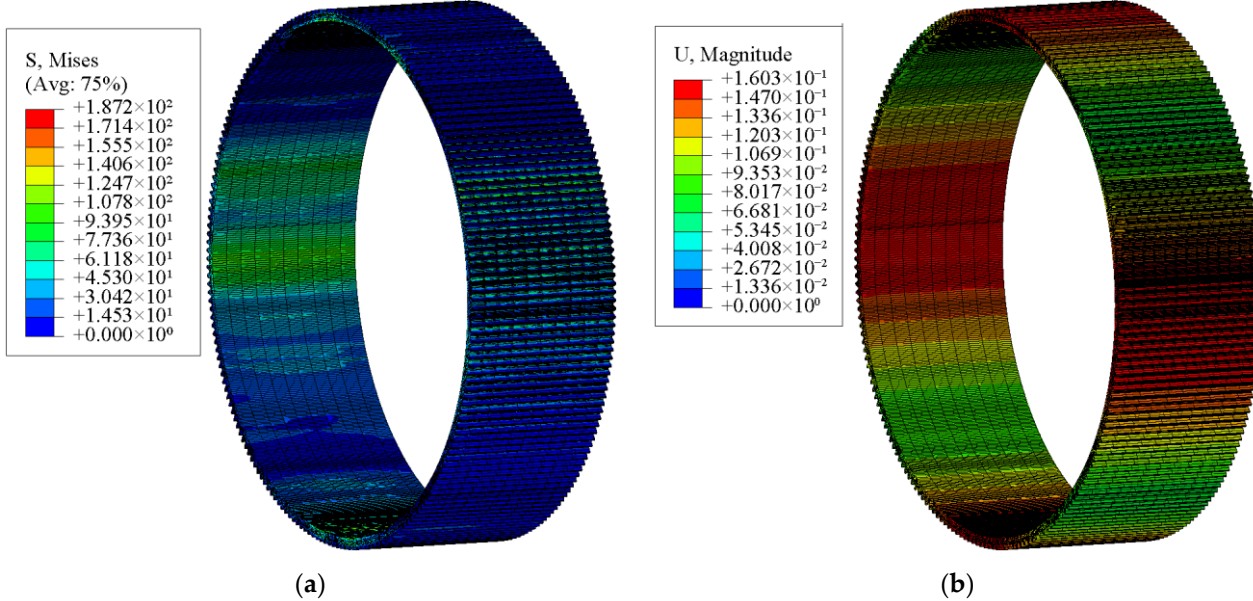

**Figure 19.** (**a**) Equivalent stress of the flexible wheel after assembly; (**b**) equivalent deformation of the flexible wheel after assembly.

It can be seen that the maximum stress and deformation of the flexible wheel under no-load after assembly occur at the ends of the long and short axes of the wave generator. The equivalent stress and deformation of the two gears under full load are shown in Figures 20 and 21, respectively. According to Figure 20, the maximum stress of the disposable flexible wheel did not reach the yield strength. Therefore, the disposable HD can meet the requirements of short-term operation under a full load. The maximum stresses of the two gears both appeared in the middle of the contact area. The deformation of the flexible wheel in the contact area slightly decreased and then gradually increased,

which is consistent with the trend of the theoretical results in Figure 15. The maximum deformation of the rigid wheel occurred in the middle position, like the stress. Additionally, the maximum stress and deformation of the flexible wheel were higher than those of the rigid wheel. Then, the load and comprehensive displacement curves of each contact tooth pair were extracted, as shown in Figure 22.

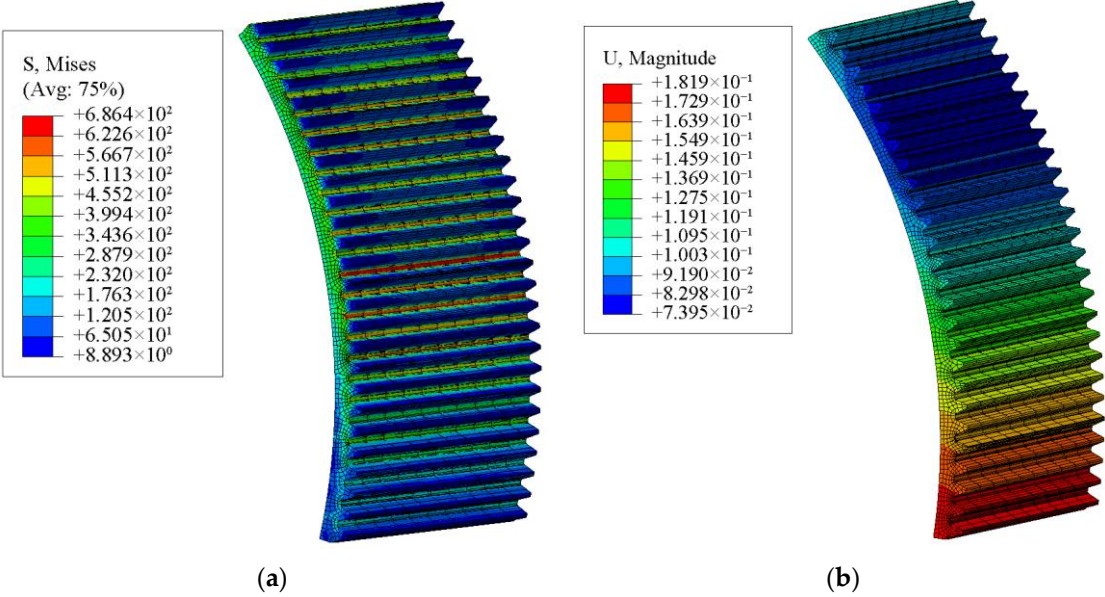

(**a**) (**b**)

**Figure 20.** (**a**) Equivalent stress of the flexible wheel after loading; (**b**) equivalent deformation of the flexible wheel after loading.

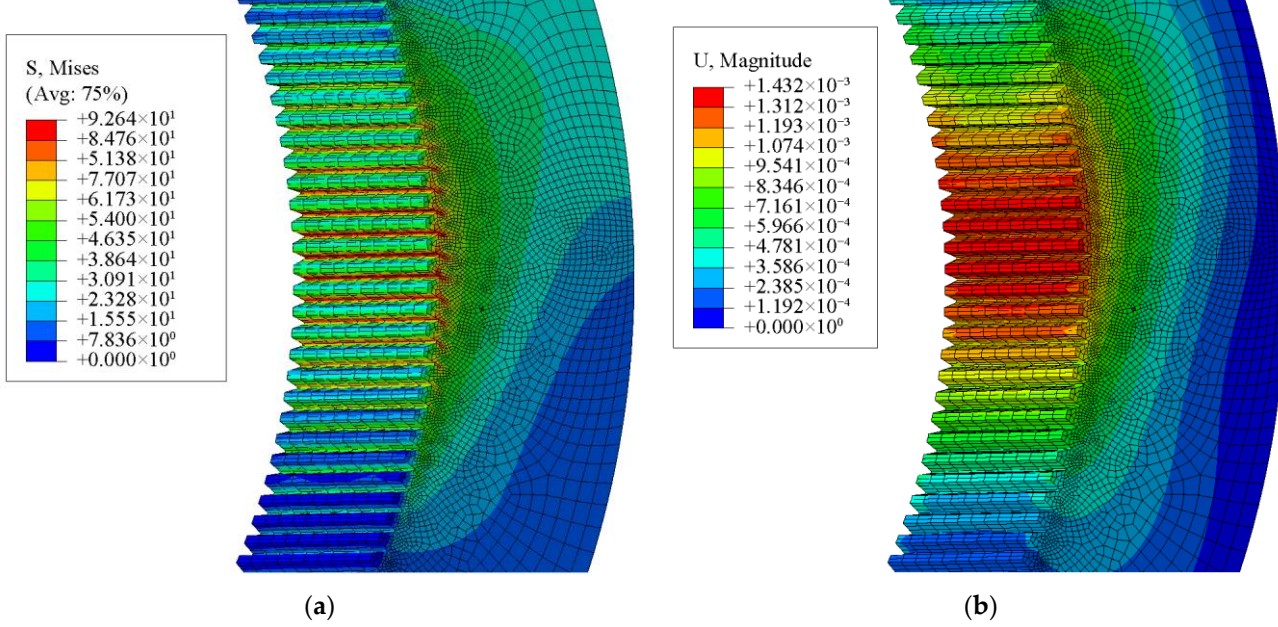

(**a**) (**b**)

**Figure 21.** (**a**) Equivalent stress of the rigid wheel after loading; (**b**) equivalent deformation of the rigid wheel after loading.

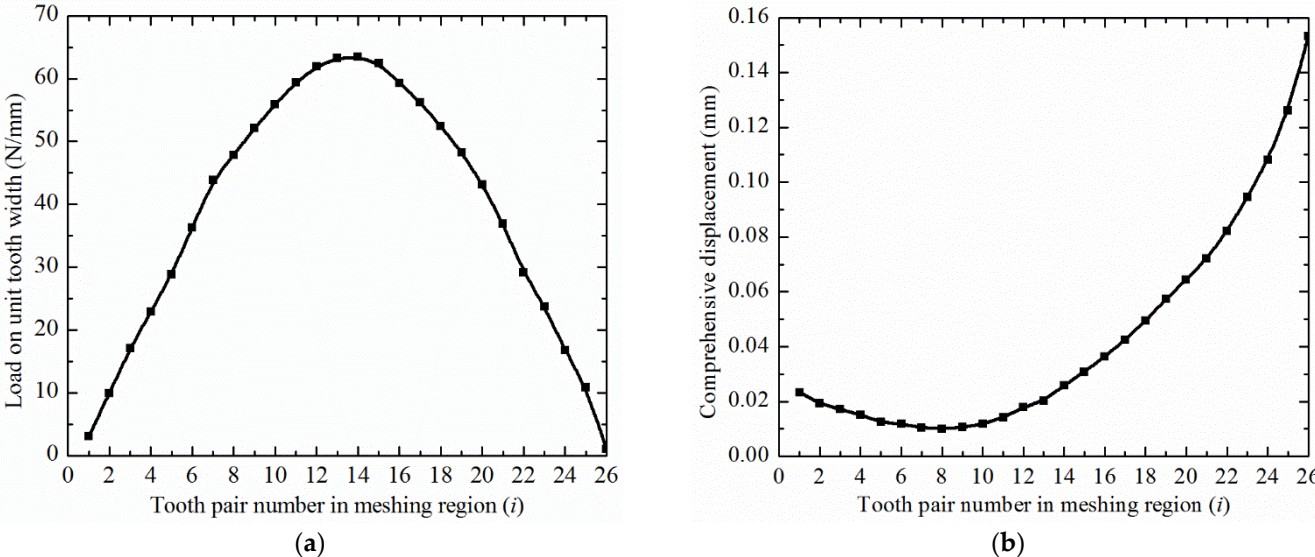

**Figure 22.** (**a**) Load on the width of the unit tooth of each node in the meshing region; (**b**) comprehensive displacement of each node in the meshing region.

As shown in Figure 22, during one engaging-in and engaging-out cycle of a tooth pair in the meshing region of the disposable HD, the load on the teeth of the gear gradually increased to the peak value and then reduced. Additionally, the load peak was distributed in the middle gear teeth in the meshing region. With subsequent teeth meshing, the previous meshing teeth did not withdraw. Thus, the superposition of the elastic displacement of the teeth of the gear led to a gradual increase in the comprehensive displacement. The results of the modified analytical method described in Section 3 and the FEM were then compared (see Figure 23).

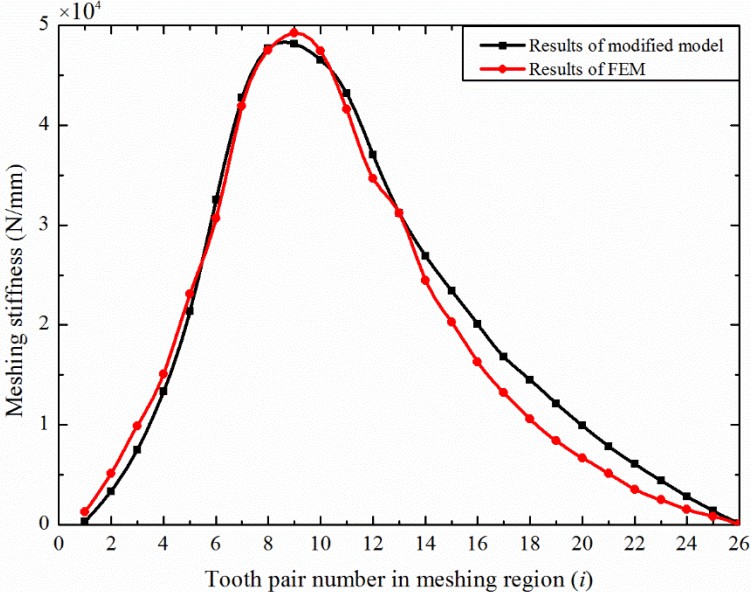

**Figure 23.** Coefficients of stiffness obtained by the modified analytic method and the FEM.

According to Figure 23, at the beginning of meshing, the stiffness of the teeth of the gear increased rapidly and then decreased gradually. Considering the influence of multi-tooth meshing on the meshing stiffness of the disposable harmonic drive, the peak value and trend of the stiffness curve obtained by the modified analytical method and FEM were

very close, which confirms the feasibility of the modified analytical method in calculating the meshing stiffness of the disposable harmonic gear under full load.

According to Equation (48), the comprehensive meshing stiffness of the disposable harmonic drive was approximately a straight line (see Figure 24). The curve in the range of ordinate 0–5 in Figure 24 is the superposition of the meshing stiffness in Figure 23. The simultaneous contact of multiple pairs of gear teeth during transmission could ensure the stability of the disposable harmonic drive. In addition, the comprehensive meshing stiffness of the disposable harmonic drive was higher than that of conventional gear, which also ensured the possibility of the disposable harmonic drive achieving a short-term full load or overload transmission.

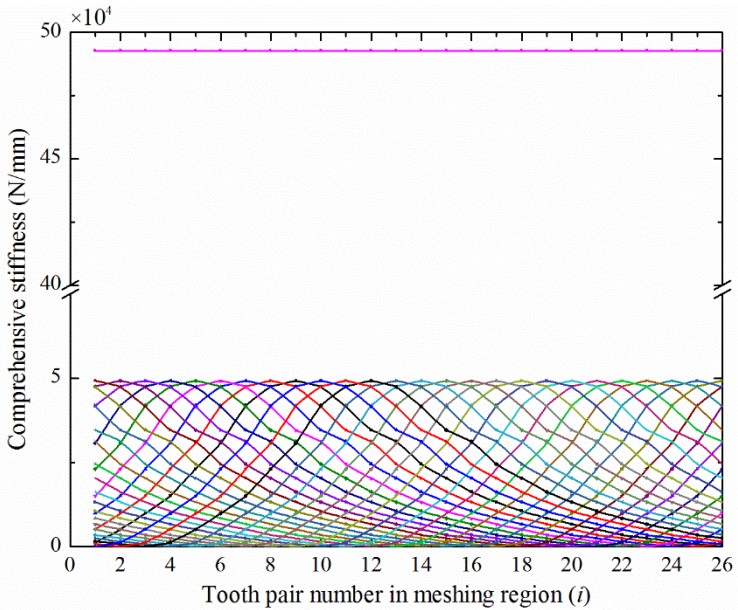

**Figure 24.** Comprehensive meshing stiffness of the disposable HD.

To simulate the failure mode of the disposable harmonic gear in case of overload, a torque of 100 N·m was applied, and the equivalent stress and plastic deformation of the disposable flexible wheel were obtained according to the material properties in Figure 3.

Figure 25a shows the stress of the flexible wheel under overload. The root stresses of several flexible teeth were higher than the allowable value of the material, and the inner wall of the flexible wheel was overstressed. Figure 25b shows that plastic deformation would occur at the root of flexible teeth under overload. The load and comprehensive deformation of the meshing tooth pair under overload were extracted. It can be seen from Figure 26a that the stress of the eleventh pair of the meshing teeth exceeded the limit value of the material. Additionally, Figure 26b shows that the subsequent tooth pairs had obvious distortion. According to Figure 26, the curve of the stiffness before damage to the flexible wheel is shown in Figure 27.

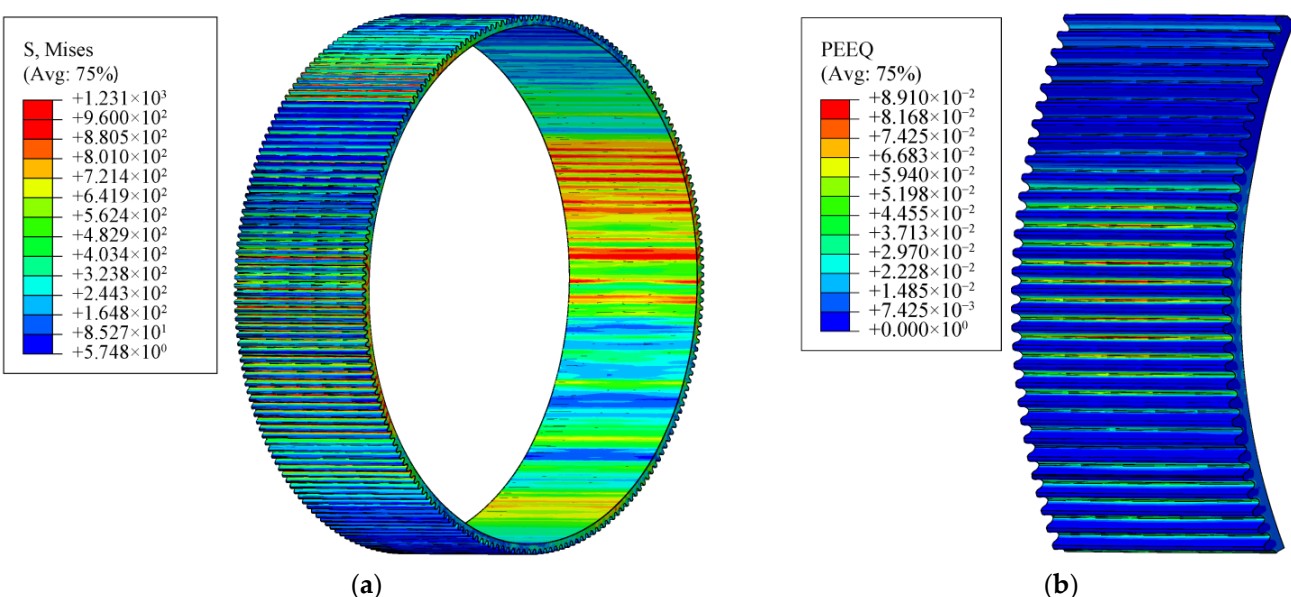

**Figure 25.** (**a**) Equivalent stress of the flexible wheel in case of overload; (**b**) plastic strain of the flexible wheel in case of overload.

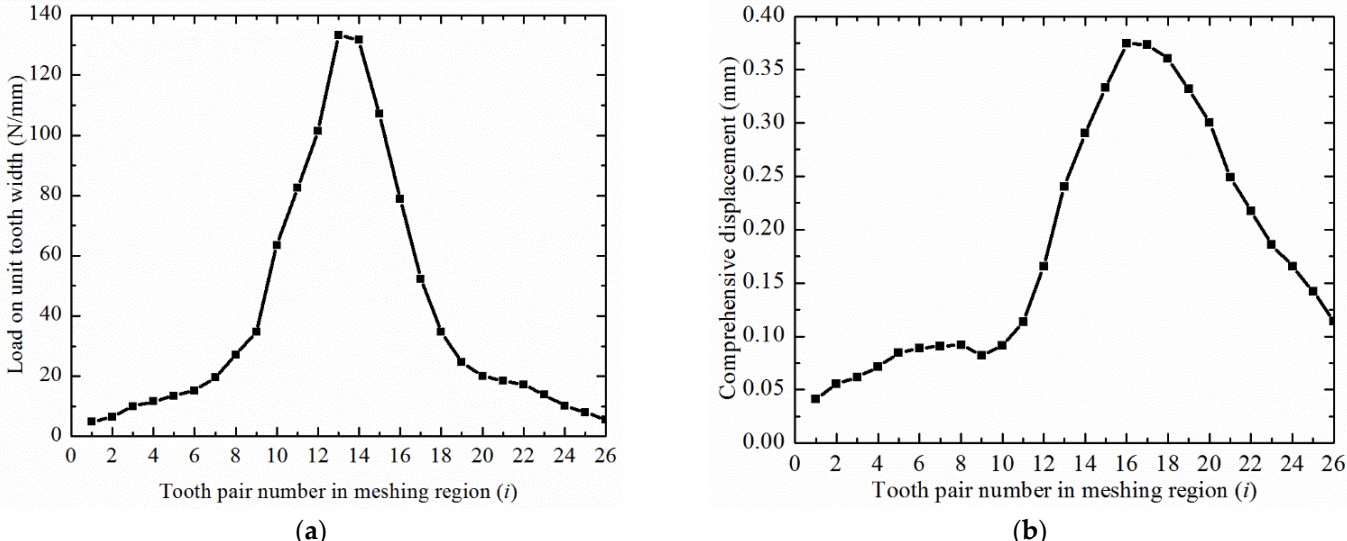

**Figure 26.** (**a**) Load on the unit tooth width of each node in the meshing region; (**b**) comprehensive displacement of each node in meshing region in case of overload.

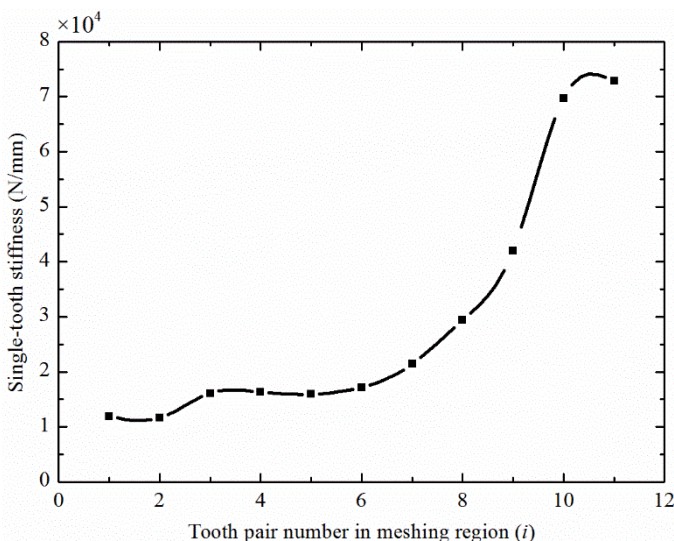

**Figure 27.** Comprehensive meshing stiffness of disposable HD.

### 5. Conclusions

To study the meshing stiffness of a disposable harmonic gear under a full load, a modified improved energy method and a FEM were proposed in this study. Compared with a conventional HD, a disposable HD is significantly different in terms of the application environment and flexible wheel structure. The stiffness of the flexible gear was calculated by using the improved energy method and considering the influence of multi-tooth meshing on the deformation. The stiffness of the rigid wheel was decomposed into bending stiffness, shear stiffness, compression stiffness, and gear foundation stiffness. Then, a comprehensive stiffness model of multi-tooth meshing of disposable HD was established. Finally, the FEM was established to verify the accuracy of the analytical model and analyze the failure form of the disposable HD under overload. The conclusions of this work can be summarized as follows:

(1) Different from other gear transmissions, the calculation of disposable harmonic gears needs to be conducted separately by distinguishing the structural characteristics of the two gears. The model of the teeth that considers the thin rim of the flexible wheel can accurately describe the amplitude of the meshing stiffness of the disposable harmonic gear under full load;

(2) The modified improved energy method considers the influence of multi-tooth meshing on the stiffness of the flexible gear and can accurately reflect the comprehensive stiffness of the disposable harmonic gear in the meshing region under full load;

(3) The comprehensive stiffness of the disposable harmonic drive is higher than that of conventional gear drive. The disposable harmonic gear can operate under full load for a short time.

**Author Contributions:** Conceptualization, G.W. (Guanglin Wang), Y.L. and X.P.; methodology, Y.Z.; software, Y.Z.; validation, Y.Z. and Y.L.; formal analysis, G.W. (Guanglin Wang); writing—original draft preparation, Y.Z.; writing—review and editing, X.P.; visualization, G.W. (Guanglin Wang) and G.W. (Guicheng Wu); funding acquisition, Y.L. All authors have read and agreed to the published version of the manuscript.

**Funding:** This research was funded by the National Natural Science Foundation of China, grant number 51875117, and the Fundamental Research Funds for the Central Universities, grant number LH2020E035.

**Institutional Review Board Statement:** Not applicable.

**Informed Consent Statement:** Not applicable.

**Data Availability Statement:** The data presented in this study are available upon request from the corresponding author. The data are not publicly available due to copyright issues.

**Conflicts of Interest:** The authors declare no conflict of interest.

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
