# Peer review of "Meshing Stiffness Calculation of Disposable Harmonic Drive under Full Load"

_machines, doi:10.3390/machines10040271_

Round 1

Reviewer 1 Report

I ask the authors of the article to respond to the following comments:

1. Equations 1 to 9 describe Rigid tooth profile fitted by envelope method resulting in Equation 9 – yg(x)=… . However in Analytical Model to Compute Meshing Stiffness was used the equivalent model based in involute tooth profile. Explain where the equation for the envelope method (Equation 9) is then located in the equations for the Meshing Stiffness model.

If not, what will be the correlation between the results of the involute model used and the true (real) tooth profile created by the envelope method typical for HD gearboxes?

2. The FS is usually ended by a flange from which the output torque is taken. However, the results of the FEM analysis show that the model had no flange, shape of FS was even completely straight. The FS did not have an internal shoulder as shown in Figure 2. Whereas in reality the deformation of Flexible Spline is completely different from the results presented in this paper. To what extent are the presented results credible?

3. On page 22 of the paper and in figure 25b the Plastic Strain was evaluated. But only the linear material model for FEM analysis was described on page 17. How was the plastic strain calculated and evaluated?

4. The introduction part should be definitely broaden. The described methods should be also confronted and compared with the study

Global Sensitivity Analysis of Chosen Harmonic Drive Parameters Affecting Its Lost Motion by Hrcek et al.

formal comments:

  • Figure 6 curve profile of cam generator - a suitable scale could be used to display the profile curve,
  • Figure 24 Comprehensive meshing stiffness of disposable HD - inappropriate y-axis scale, in range 0-5 poorly visible curves,

I recommend editing these images

Reviewer 2 Report

  1. Even with Disposable Harmonic Drive, I think it is necessary to mention the effect of shortening the lifespan and generating errors due to problems such as friction/wear/lubrication.
  2. In the case of Figure 14 on page 14, explain Figure.s 14 (a) and (b) separately in the text.
  3.  Briefly explain the explanations from Page 18 to Figure 18 to Figure 22 by dividing them into (a) and (b) in the text.
  4. Modify Figure 26 to Figure 25 on Page 22.
  5. Divide Figures 25-26 into (a) and (b) in the text and explain.
